# Manipulating Topological Phases in Magnetic Topological Insulators

**DOI:** 10.3390/nano13192655

**Published:** 2023-09-27

**Authors:** Gang Qiu, Hung-Yu Yang, Su Kong Chong, Yang Cheng, Lixuan Tai, Kang L. Wang

**Affiliations:** 1Department of Electrical and Computer Engineering, University of California, Los Angeles, CA 90095, USA; hungyuyang@g.ucla.edu (H.-Y.Y.); sukongc@g.ucla.edu (S.K.C.); cheng991@g.ucla.edu (Y.C.); lixuantai@ucla.edu (L.T.); 2Department of Electrical and Computer Engineering, University of Minnesota, Minneapolis, MN 55455, USA; 3Beijing Academy of Quantum Information Sciences, Beijing 100193, China

**Keywords:** magnetic topological insulator, quantum anomalous Hall effect, topological phase transition

## Abstract

Magnetic topological insulators (MTIs) are a group of materials that feature topological band structures with concurrent magnetism, which can offer new opportunities for technological advancements in various applications, such as spintronics and quantum computing. The combination of topology and magnetism introduces a rich spectrum of topological phases in MTIs, which can be controllably manipulated by tuning material parameters such as doping profiles, interfacial proximity effect, or external conditions such as pressure and electric field. In this paper, we first review the mainstream MTI material platforms where the quantum anomalous Hall effect can be achieved, along with other exotic topological phases in MTIs. We then focus on highlighting recent developments in modulating topological properties in MTI with finite-size limit, pressure, electric field, and magnetic proximity effect. The manipulation of topological phases in MTIs provides an exciting avenue for advancing both fundamental research and practical applications. As this field continues to develop, further investigations into the interplay between topology and magnetism in MTIs will undoubtedly pave the way for innovative breakthroughs in the fundamental understanding of topological physics as well as practical applications.

## 1. Introduction

The discovery of the quantum Hall effect (QHE) marked the commencement of a new era of topology in condensed matter physics [1]. In 1980, von Klitzing et al. reported that under a strong perpendicular magnetic field, the Hall conductance of a two-dimensional electron gas system only takes a set of values in the unit of a universal quantum conductance e^2^/h, which is independent of any material-specific properties [1]. At the nascence of QHE, its profound connection with the concept of topology in mathematics was yet to be revealed (though the picture of zero-resistance edge states and insulating bulk states did give a hint). Shortly after, theorists worked out that when two-dimensional electron gases are subject to magnetic fields and periodic potentials, the filling factor υ of the QHE states is equivalent to the topological invariant [2,3], now known as the Thouless–Kohmoto–Nightingale–Nijs (TKNN) number or the Chern number. These topological invariants remain unchanged under smooth deformation, which gives rise to the robustness of quantization in the QHE. Since then, the concept of topological classification has quickly expanded to other gapped electronic systems, leading to the discovery of topological insulators. Instead of creating topological energy gaps with Landau levels under a magnetic field, topological phases can exist in certain insulators with inverted conduction and valence band with strong spin–orbit coupling. One prominent example of topological insulators in the three-dimensional (3D) limit is a family of tetradymite compounds, such as Bi_2_Te_3_, Sb_2_Te_3_, and Bi_2_Se_3_ [4,5,6,7]. These 3D TIs feature inverted gap states in the bulk and gapless Dirac surface states [5]. In the 2D limit, the 2D TIs have been theoretically proposed [8,9] and experimentally reported in HgTe quantum wells [10]. These 2D TIs, also known as quantum spin Hall (QSH) insulators, host two counter-propagating edge states with opposite spins, which draws an analogy to the QHE but with the time-reversal symmetry preserved. Despite the topological nature, the counter-propagating helical edge states are still susceptible to spin-flipping scattering processes, which limit the scale of QSH to extremely short distances and hinders the direct observation of the QSH effect.

Magic happens when magnetic ordering is introduced into 2D or 3D TIs. By adding magnetization in the out-of-plane direction, the exchange interaction opens up a gap at the surface Dirac states, and the metallic surface becomes insulating. While the gapless feature is still preserved at the edge of the film, the degeneracy protected by the time-reversal symmetry is lifted, leading to a chiral edge current with a unidirectional flow of spin current, the direction of which depends on the magnetization. This effect is called the quantum anomalous Hall effect [11,12,13], which can be traced back to Duncan Haldane in 1988 [14]. By applying a periodic magnetic field (but with a zero net field), zero-field quantization can be achieved in a honeycomb lattice [14]. Although this approach is challenging to implement in practice, it prophesies QAH states without a magnetic field. In 2008, QAH states were predicted in a Mn-doped 2D TI HgTe quantum well [15]. However, in this system, the magnetization is mediated by the Ruderman–Kittel–Kasuya–Yosida (RKKY) mechanism from the itinerant electron bands, which will be suppressed in the insulating states. Later, an alternative approach was proposed in Cr or Fe-doped 3D TI thin films [16], from which Cr doping was proven to be feasible and led to the experimental realization of QAH states.

While the advancement in material design and optimization has led to incremental improvement in QAH temperatures, manipulating MTIs in other parameter spaces is still urgently needed toward higher-temperature QAH effect and other exotic physics in novel topological phases. With a few recent outstanding review papers on MTIs and the QAH effect [11,12,13,17,18,19,20,21], in this paper, we turn our focus to discussing how to manipulate topological phases in MTIs in a controllable manner by tuning external conditions such as pressure, electric field, and interfacial proximity effect. This review paper is structured in the following way: in the rest of Section 1, we introduce the MTI material categories and different topological phases associated with them; in Section 2, we discuss the scaling behavior during quantum (topological) phase transitions in finite-sized MTI structures; in Section 3, we address how the band topology and magnetic properties of MTIs evolve under pressure; in Section 4, we review the impact of electric field on modifying the band topology; Section 5 focuses on the proximity effect as a promising approach to combine topological materials with magnetic materials; we conclude in Section 6 with outlooks for future endeavors.

### 1.1. Material Platforms

#### 1.1.1. Magnetically Doped TIs

Transition metal-doped TIs have been so far the prevailing and the most sophisticated platform to achieve the QAH effect. For the hosting TI materials, although a few telluride and selenide materials are promising candidates with inverted bands in a 3D limit, only Bi_2_Te_3_ and Sb_2_Te_3_ exhibit long-range ferromagnetic ordering when Cr is doped, whereas the other selenides show a weak superparamagnetic behavior due to aggregations of magnetic dopants [22,23]. In addition, native doping due to lattice defects tends to shift the Fermi level away from the topological gap (Bi_2_Te_3_ is typically n-type, whereas Sb_2_Te_3_ is p-type). Therefore, the QAH effect (Figure 1a) can only be achieved by carefully tuning the stoichiometric composition of Cr-(Bi,Sb)_2_Te_3_ (CBST) [24,25,26]. Later, V was used to dope (Bi,Sb)_2_Te_3_ (VBST), which shows a much harder ferromagnetic ordering with a large perpendicular magnetic anisotropy [27]. The critical temperature for the QAH effect was also slightly improved. Although tremendous effort [28,29,30] has been made toward improving VBST’s and CBST’s sample quality, the QAH effect is still only achievable at an extremely low cryogenic temperature (<2 K), which is significantly lower than the expected exchange gap energy and the Curie temperature of the ferromagnetic ordering (~25 K). The suppressed thermal activation gap is likely due to the spatially varying chemical potential in the film [31,32].

#### 1.1.2. Intrinsic Magnetic Topological Insulators

Introducing magnetism into TIs through doping inevitably deteriorates the material quality and limits the QAH temperature. Alternatively, certain materials inherently possess a non-trivial band topology and magnetic ordering, which can potentially preserve the quality of the materials and therefore achieve the QAH effect at an elevated temperature. One example is the layered A-type anti-ferromagnetic topological insulator, MnBi_2_Te_4_, as demonstrated in Figure 1b. MnBi_2_Te_4_ has a van der Waals layered structure, where each septuple layer can be viewed as a MnTe layer sandwiched by a Bi_2_Te_3_ quintuple layer. The Bi_2_Te_3_ contributes to the topological nature, and the magnetic moments of Mn atoms give rise to magnetic orderings [33,34]. The QAH effect was first reported in odd-layer MnBi_2_Te_4_ in 2020 at a slightly elevated temperature compared to CBST systems [35]. Depending on the number of layers and spin configurations, MnBi_2_Te_4_ can also host a variety of topological phases including QAH, Chern insulators, Weyl semimetals, and axion insulator states, which will be further elaborated below.

#### 1.1.3. Twisted Moiré Materials

By twisting 2D materials (either homogenously or heterogeneously), Moiré superlattice can be created with a superlattice periodicity varying in a wide range, typically much larger than the lattice constant. This allows a rich phase diagram in these twistronics with strong correlations and/or breaking symmetries. In order to observe QAH states in twisted Moiré materials, breaking time-reversal symmetry is required. This can be achieved with orbital magnetization in twisted bilayer graphene on top of aligned hexagonal boron nitride, where orbital ferromagnetism is manifested by the observation of anomalous Hall effect [36]. Shortly after, the QAH effect was observed by improving the material quality and fabrication procedures [37]. A higher Chern number as well as fractional Chern insulating states were also observed in different Moiré band filling factors [38,39,40,41]. Recently, QAH states have also been reported in transition metal dichalcogenide heterostructures as depicted in Figure 1c [42]. 

### 1.2. Topological States in Magnetic Topological Insulators

Beyond QAH states, MTIs can also host or are closely related to other topological phases of matter through topological phase transitions, including: axion insulators, Weyl semimetals, high-Chern-number insulators, and higher-order topological states, just to name a few. Here, we provide an overview of a few representative examples of topological states that are accessible by tuning material parameters of MTI within an experimentally feasible range.

#### 1.2.1. Axion Insulators 

In an axion insulator, the electromagnetic response acquires a coupling between the **E** and **B** fields in terms of **E∙B**, resembling the axion field in particle physics [43]. Such a magnetoelectric effect enabled by non-trivial topology will lead to, e.g., a quantized polarization response when a magnetic field is applied, or vice versa [44,45]. A material platform for axion insulators starts from an MTI, but instead of having the same magnetic layer at the top and bottom surfaces, magnetic layers with different coercive fields are used for the top and bottom surfaces [44]. As a result, there exists a range of magnetic field where the top and bottom magnetizations point in opposite directions, leading to an axion insulator, as shown in Figure 1d. Such a material platform has been investigated by combining Cr-doped and V-doped MTIs, and signatures of axion insulators such as zero-Hall plateaus distinct from those in a trivial insulating transition state in CBST have been observed [46,47,48]. It remains an active research area to directly observe quantized topological magnetoelectric effect in axion insulators.

#### 1.2.2. Chern Insulators with High Chern Numbers 

The first-ever discovered QAH insulator was in CBST where the entire thin film was uniformly doped [24,25] (Figure 1a). Later, it was realized that one only needs to make sure both top and bottom surfaces are magnetized, and a heterostructure with MTI/TI/MTI modulation doping can have an even higher onset temperature of QAHE compared to uniformly doped MTI [49]. In all the above cases, there is only one pair of top and bottom surfaces, and thus one copy of the chiral edge state which leads to Chern number C = 1. However, by inserting extra MTI/TI layers into the heterostructure, multiple copies of paired surfaces can coexist in one single heterostructure and add up to higher integers of Chern numbers; for example, Figure 1e illustrates an MTI/TI/MTI/TI/MTI heterostructure with a Chern number C = 2, as can be seen in its transport properties [50,51]. The realization of such stacks provides arbitrary tunability of Chern number and chiral edge states.

#### 1.2.3. Magnetic Weyl Semimetals

Three-dimensional Weyl semimetals host topological Weyl nodes of opposite chirality in their band structure, created by breaking time-reversal symmetry and/or inversion symmetry (see Figure 1f) [52]. These topological nodes are connected by edge states on the surface called Fermi arcs, guaranteed by the topological band structure [53]. In magnetic Weyl semimetals, as the dimension is reduced and the system becomes 2D, the Weyl nodes may annihilate on the edge of the Brillouin zone, and the Fermi arcs can evolve into chiral edge states, leading to a QAH insulator [54,55]. The connection between magnetic Weyl semimetals has been discussed in theory [54,55] while the experimental demonstration still awaits.

## 2. Scaling Behavior during Quantum Phase Transitions in Finite-Sized MTIs

We first discuss the finite size effects on the transport behavior of mesoscopic MTI devices, particularly during their topological phase transition regime. In general, the topological behavior of MTIs is universal and should be independent of the geometry of the sample. However, when the physical size of the device is below a certain characteristic length of the electron quantum transport, (i.e., phase coherence length), the electron transport behavior quickly deviates from the classical diffusive regime and should be characterized by coherent mesoscopic transport. Understanding the behavior in the mesoscopic limit becomes especially important as the QAH effect holds great promises to be implemented in future dissipation-less, field-free quantum devices. 

A key question to be addressed is to determine the minimum size limit below which the QAH effect breaks down. Recent work has demonstrated that the quantization is well preserved in the deep sub-micrometer size limit [56,57,58]. For example, when the MTI films were patterned into miniaturized Hall bar devices with a sub-µm channel width as shown in Figure 2a, the magnetic hysteresis loop of $\rho_{xx}$ and $\rho_{xy}$ still resembled the behavior in large-scale devices (Figure 2b,c) [56]. This quantization can be observed in devices down to 72 nm, suggesting the decaying length of the chiral edge states is below 36 nm [57], which is significantly narrower than the chiral edge states in quantum Hall systems [59,60,61]. This discrepancy arises from different mesoscopic origins of chiral edge states in QAH and QH effects. In QH systems, the chiral edge states are formed by alternating compressible and incompressible strips, and the overall width of the edge states (typically on the order of 1 µm) is determined by the carrier density, filling factors, and dielectric constant [59,60,61]. In contrast, the bulk of QAH films are carrier-free, and the width of the chiral edge states is solely dependent on how wide the wave function of the edge modes stretches, i.e., the Fermi wavelength of the chiral edge states [56]. By measuring the temperature, current bias, and gate dependence, the breakdown of the QAH effect in finite-sized MTI samples can be attributed to bulk localized states mediated by the cross-channel back-scattering mechanism [56], as demonstrated in Figure 2d,e. In addition to quantized states, during the magnetic switching, telegraph noises due to magnetic fluctuations are also observed in sub-µm devices, and careful analyses of these noise spectrum give an estimated magnetic domain size of an ~100–200 nm range [56,58]. 

Whilst the quantization is persistent in quantized states in mesoscopic MTI devices, the scaling behavior shows a strong size dependence in the quantum critical regime during the topological phase transition [62]. A typical topological phase transition between the QAH, insulating, and quantum critical regimes is depicted in Figure 3a–d. By controlling the magnetic field, the MTI films can be continuously tuned from trivial insulating to QAH insulating states, bridged by a quantum critical regime. The scaling behavior of the inelastic scattering length follows a power law as a function of temperature, and a temperature exponent coefficient can be extracted by measuring temperature or current bias dependence in different quantum phases. Here the inelastic scattering length describes the average distance between inelastic scattering events, and it dictates whether a device is in a classical diffusive limit or quantum coherent transport. It has been shown that the temperature exponents remain a universal value in the trivial insulating, quantum critical, and QAH insulating regimes, indicating the nature of the interactions between electrons, and disorders remain in the same mechanism throughout the topological phase transitions [62]. It should be noted that when the dimension of the sample is smaller than the inelastic scattering length, while the sample can still reach quantization, the scaling behavior during the topological phase transition starts to deviate from a universal power law. Since the inelastic scattering length scales with the temperature exponentially, we observe a finite-size effect in the scaling behavior at low temperatures where the inelastic scattering length exceeds the sample size (Figure 3e). The saturation temperature at which the device size takes over can be used to determine the inelastic scattering length (Figure 3f,g), which is in the order of 1 µm at 200 mK [62]. 

## 3. Pressure Tuning of Topological Phases

The quest for tuning the properties of MTI to realize the QAH effect at a favorably high temperature emerged as an increasingly important research topic in condensed matter physics in the past decade. Unfortunately, the disorder in MTI caused by the random distribution of magnetic dopants limits the quantized phenomenon to extreme cryogenic conditions, restricting their practicality. While sample properties can be controlled via chemical substitution, in recent years only minor improvements in the operating temperature have been realized through this route, demanding alternative axes along which the QAH effect may be tuned. Lattice distortion induced by pressure is a common technique to alter the band structure and induce topological phase transitions. Topological phase transitions under pressure have been widely reported in various topological materials [63,64,65,66,67,68,69]. In the case of MTIs, the pressure can have a substantial impact not only on electronic structures [68,70] and spin-orbit coupling strength, but also on magnetic structures [71,72,73,74,75,76]. Intuitively, reduced interatomic distance under pressure can potentially enhance the magnetic exchange interaction, and pressurizing MTI will consequently increase the magnetism. Hence, pressure is an effective route to shift between topological phases in MTIs by manipulating the interplay between topology and magnetism. In this section, we review recent reported pressure-tuned topological phase transitions in MTI systems. 

The transport behavior in quantized CBST has been systematically investigated in response to continuous pressure tuning, achieved via the application of hydrostatic pressure using a piston cell with a tri-axial compressive strain (Figure 4a,b) [77]. Under a modest pressure of up to 1.6 GPa, the shrinking unit cell size leads to a reduced topological gap, as evidenced by a deviation from the quantization in both gate voltage traces (Figure 4c,d) and field hysteresis loops (Figure 4e,f). In the meantime, the coercive field of ferromagnetic states increases under pressure, suggesting magnetic order is enhanced (Figure 4g). This pressure-driven enhancement of magnetic order can be simply understood by the reduction of lattice constants leading to stronger exchange interactions. However, the enhanced magnetic ordering should lead to an expanding exchange gap, which seems to oppose the observed topological gap closing in transport measurements. Therefore, it is speculated that other mechanisms should be accounted for in the suppressed thermal activation gap such as surface hybridization or occupation of delocalized states. To understand the mechanism of topological phase transition under pressure, first-principles calculations were performed, and a rich topological phase space was revealed. While both hybridization and exchange gap will expand under pressure, the competition between these two leads to a transition between QAH insulator and trivial insulator states (Figure 4h). In addition, the valence band maxima along the Γ-M line also shift upwards, leading toward a metallic phase, which can explain the gap-closing trend observed in experiments. 

These findings demonstrate that the band structure of magnetically doped (Bi_1-x_Sb_x_)_2_Te_3_ may be directly and predictably addressed through structural tuning. A critical pressure of ~3 GPa is extrapolated for a QAH-insulator-to-metal transition. While previous experiments were performed with compressive strain, it is foreseeable that a tensile strain may lead to an expansion of the QAH gap and enhance the critical temperature. In addition, a structural transition from rhombohedral to monoclinic crystalline structures has been predicted at about 9 GPa, which may lead to even richer topological phase transitions [65,78,79]. These findings establish strain engineering, a heretofore unexplored avenue, to be a promising route toward tailored electronic and magnetic properties in wafer-scale CBST-based QAH materials in the future. 

In addition to CBST, a pressure-dependent transport study was also recently reported in MBT films [80]. With the layered A-type AFM magnetic configuration and relatively weak van der Waals interlayer interactions, the interlayer exchange couplings should be greatly enhanced under pressure, which will significantly alter the magnetic and topological phases in MBT. In particular, because even- and odd-layered MBT films have different compensated/uncompensated net magnetizations, their ground states’ evolution under pressure also differs drastically. In an even-layered MBT with fully compensated magnetization, Chern insulating states can be achieved even in spin-flop states at a moderate magnetic field of 2.5 T, which is significantly lower compared to the onset of FM states (~5 T) under ambient pressure, whereas an odd-layered MBT at a zero field will undergo a phase transition from a trivial insulator to a QAH insulator by tuning the pressure [80]. 

## 4. Electric Field Tuning of Topological Phases

### 4.1. Theoretical Models

In thin films of MTIs, the interaction between the surface hybridization gap ($m_{0}$) and the magnetic exchange gap (Δ) gives rise to distinct topological phases. Applying an electric field is equivalent to changing the potential difference (V) between the two surfaces. Based on the effective Hamiltonian model for 2D surface states [81,82], the surface wavefunctions can be modulated by V, leading to changes in the global band gap and the magnetic exchange gap. The band gap can be expressed as $E_{g}=2∆-2\sqrt{m_{0}^{2}+V^{2}}$. The gap closing point occurs at $V_{c}=\sqrt{{∆}^{2}{-m}_{0}^{2}}$, indicating a topological phase transition between the QAH insulator and normal insulator states. The topological and trivial insulator phases can be distinguished by the difference between two energy scales: Δ and $m_{0}$, where Δ > $m_{0}$ indicates the topological phase. The application of V can increase the hybridization gap, resulting in a trivial insulator phase with Δ < m^*^, where $m^{*}=\sqrt{m_{0}^{2}+V^{2}}$.

The exchange interaction between surface electrons and magnetic dopants can be described by the term $-J_{eff}\vec{S}\left({\vec{r}}_{i}\right)\cdot \vec{S}\left({\vec{r}}_{j}\right)$, where $J_{eff}\sim J^{2}\left(\frac{1}{U}-\chi_{e}\right)$ [81,83]. Here, J represents the exchange coupling parameter, U denotes the Hubbard interaction energy, and $\chi_{e}$ is the spin susceptibility. The effect of V on the magnetic properties is observed in the reduction in $\chi_{e}$ as V increases. This decrease in $\chi_{e}$ is attributed to the van Vleck type of spin susceptibility, which involves the hopping of the electron spins into the itinerant bands. Moreover, the magnetic anisotropy also decreases with increasing V, as indicated by the decrease in the ratio of susceptibility tensor $\chi_{e}^{zz}/\chi_{e}^{xx}$. A magnetic phase transition can occur from a ferromagnetic phase to a paramagnetic phase when the spin susceptibility becomes smaller than 1/U.

### 4.2. Experimental Observations

The effect of electric fields on MTIs can be categorized into two aspects: electric field-induced shifts in band structures [81,84], and electric field-controlled magnetic properties [85,86]. We will discuss the two types of electric field-controlled quantum phase transitions in MTIs: magnetic phase transition and topological phase transition.

#### 4.2.1. Magnetic Phase Transition

Zhang et al. [85] demonstrated the experimental observation of electric field-controlled magnetic phase transition in CBST. By adjusting the composition close to the topological quantum critical point, the application of a gate electric field effect induced a magnetic phase transition from a ferromagnetic state to a paramagnetic state. Figure 5b shows the anomalous Hall effect in 8QL of CBST measured at different gate voltages ranging from gate voltage (V_g_) of −210 to 210 V. The hysteresis loop disappeared at V_g_ = −25 V, and the anomalous Hall curves reversed sign with a negative weak-field curve, characteristic of a paramagnetic state. The gate voltage has two effects: changing the carrier density and applying the electric field. The modulation of the magnetic properties through the change in carrier density can occur via the surface Dirac fermion-mediated Ruderman–Kittel–Kasuya–Yosida (RKKY) mechanism [87,88]. However, in this case, the studied material is heavily n-type doped with an electron carrier density of 10^14^/cm^2^, while the solid-state gating typically changes carrier density at most on the order of 10^13^/cm^2^, resulting in a relatively small change in carrier density. Additionally, the nearly constant slope of the ordinary Hall effect in R_yx_ curves at different values of V_g_ in the saturation region (Figure 5b) indicates no significant change in carrier density, thereby excluding a carrier dependent magnetic phase transition. Hence, the applied gate voltage primarily affects the electric field.

The mechanism of the magnetic phase transition is attributed to the Stark effect, where the energy levels between the p_z_ orbits of Bi and Se/Te shift, leading to a topological phase transition. This subsequently weakens the van Vleck interaction [89], which is an inter-band process between conduction and valence bands, resulting in the transition from a ferromagnetic to a paramagnetic state. The calculated spin susceptibility (Figure 5a) decreasing with electric field further supports the analyses.
Figure 5**Electric field tuning of topological phases in MTIs.** (**a**) The calculated spin susceptibility of the Cr-doped Bi_2_(Se_x_Te_1−x_)_3_ as a function of electric field. Insets show the anomalous Hall loop in magnetic field for the magnetic phase transition from ferromagnetism to paramagnetism. (**b**) Magnetic field-dependent anomalous Hall curves for 8QL (Cr_0.11_Bi_0.89_)_2_(Se_0.67_Te_0.33_)_3_ measured at different gate voltages at T = 1.5 K. (**c**) (left) DFT calculated band structures for 5QL CBST under external electric field of 0 and 1 V/nm. (Right) Schematic of dual-gated device structure under the application of electric fields to control between topological “on” and “off” states. Red arrows represent chiral edge state current directions. (**d**) Color map of longitudinal resistance (R_xx_) as a function of dual-gate voltages measured at T = 0.2 K. The temperature dependent longitudinal resistivity (R_xx_) curves measured under different electric fields as indicated by points in the color map. Figure 5a,b are adapted from ref. [85]; Figure 4c,d from ref. [90].
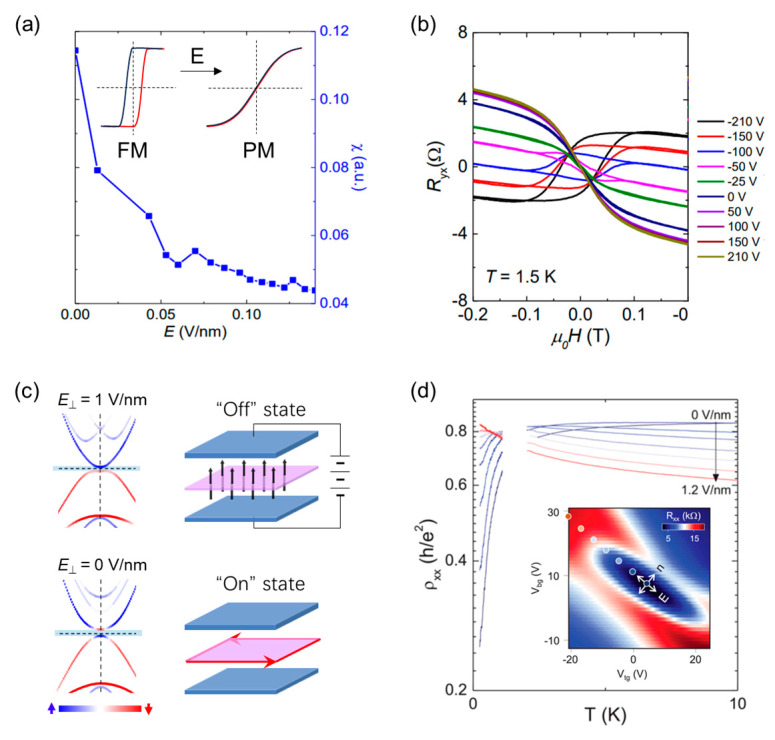


#### 4.2.2. Topological Phase Transition

Using a single gate approach may not be sufficient for tuning the QAH phase, as it tends to primarily modulate the carrier density, especially when the carrier density is low (which is a requirement for achieving a QAH state with an in-gap Chern number of 1). To overcome this limit, a dual-gating platform is needed, which enables an independent control of both the carrier density and the electric field. Fortunately, recent advancements in van der Waals stacking of two-dimensional (2D) layered materials have provided an excellent opportunity for achieving high-quality gate control using atomically flat dielectric materials. By utilizing this approach, a dual-gated device based on CBST can be realized. This is achieved by combining molecular beam epitaxy (MBE) growth on a layered mica substrate with 2D exfoliation and transfer methods [90].

Figure 5d illustrates the color map of R_xx_ and R_xy_ as a function of the top gate and back gate voltages for the 5QL CBST dual-gated device measured at 0.2K. The effect of the electric field on the QAH state is evaluated along the electric field axis in the color maps. The minimum R_xx_ gradually changes to the maximum as the electric field increases, indicating a transition into a topologically trivial anomalous Hall insulator phase [91]. The transition is evaluated by the temperature-dependent ρ_xx_ curves taken at the different electric fields, as indicated in the R_xx_ color map. The temperature profiles of ρ_xx_ exhibit a crossover from metallic behavior to weakly insulating behavior with an increasing electric field. The effective distinction between carrier density and electric field excludes the possibility of a metal-to-insulator transition resulting from the localization-induced phase transition [92], which typically arises from disorder-induced charge states localized at the hole band edge.

The mechanism of the topological phase transition can be understood as the band closing and reopening modulated by an electric field, as illustrated by the surface band structures shown in Figure 5c. The band structure of the CBST lattice exhibits an inverted surface band gap, and the calculated in-gap Hall conductivity indicates a QAH phase with a Chern number of 1 in the absence of an electric field. The band structures before and after gap closure clearly show the change in band inversion between the spin-up and spin-down bands at the Γ point. This band inversion occurs when the electric field is large enough to reverse the spin-up band of the upper surface and the spin-down band of the lower surface, resulting in a topological phase transition from the QAH state to the anomalous Hall insulator state.

### 4.3. Electric Field-Controlled Topological Transitions

The electric field-controlled topological phase transition allows for a topological field-effect switching on the 1D helical and chiral edge channels [81,93]. The on and off states (Figure 5c) of these channels are defined as topological and trivial states, respectively. The research effort toward switching between the topologically non-trivial edge states and the trivial insulator states can lead to the development of topological transistors, as proposed in 2D topological insulators with 1D helical edge states [94,95,96]. These topological transistors utilize the quantized longitudinal/Hall conductance through topologically protected helical/chiral edge states by controlling the bulk band gap [95], offering great potential for practical applications in topological quantum electronics. Table 1 summarizes the critical electric field required for topological phase transitions in various quantum material systems. For instance, in 2D topological insulators based on InAs/GaSb, Na_3_Bi, graphene/WSe_2_, and hybridized BiSbTeSe_2_, the electric field can switch between quantum spin Hall states and trivial insulating states. In systems with chiral edge states, such as (Cr,Bi,Sb)_2_Te_3_, twisted monolayer-bilayer graphene (tMBG), and MoTe_2_/WSe_2_ Moiré semiconductor heterostructure, the electric field can be used to manipulate magnetic and topological phase transitions.

## 5. Interface Tuning and Proximity Effect

Although doped MTI is an established way to introduce magnetic order into TI, the process of doping introduces impurities into the material, leading to disorder and scattering of charge carriers [32,102]. This can degrade the material’s electronic properties and hinder the transport of topological surface states, which are crucial for its unique properties. On the other hand, the magnetic proximity effect (MPE) provides a non-invasive approach to induce a magnetic order in the TI layer through magnetic exchange coupling at the interface of TI and magnetic materials. Moreover, such heterostructures offer greater flexibility in tailoring the magnetic properties to suit specific experimental or device requirements through interface engineering. Ferro-(Ferri-)magnetic insulators (MI) such as yttrium iron garnet (YIG, Y_3_Fe_5_O_12_) or EuS are commonly used as magnetic materials due to their insulating properties. In molecular beam epitaxy (MBE)-grown TI/MI heterostructures, the signature of the proximity-induced magnetic order has been observed by the emergent anomalous Hall effect (AHE) and the suppressed weak anti-localization (WAL) effect due to the cancellation of time-reversal self-interference paths [103,104,105,106,107], as shown in Figure 6a,b. Epitaxial growth may introduce a chalcogenide-rich dead layer during the annealing of the seeding layer under Se/Te fluxes owing to a large lattice mismatch between YIG and TI [108]. X. Che et al. [109] demonstrated a TI/MI heterostructure prepared through transferring a molecular beam epitaxy (MBE)-grown Bi2Se3 film onto a yttrium iron garnet (YIG) substrate via a wet transfer technique (Figure 6c). The resulting Bi_2_Se_3_/YIG heterostructure exhibits high quality and an atomically sharp interface. MPE can also occur at a TI/antiferromagnet (AFM) interface. C. Yang et al. [110] reported MPE-induced ferromagnetism in AFM CrSe/(Bi,Sb)_2_Te_3_ (BST) heterostructures. Interestingly, only CrSe grown on top of BST with Cr termination at the CrSe interface showed the MPE, while growing BST on CrSe with Se termination yielded no evidence of an MPE. This indicates the important role of the double-exchange interactions between Cr^3+^ surface states and Cr^2+^ bulk states, which stabilizes the ferromagnetic order localized at the interface. Although the MPE is believed to be a promising approach toward high-temperature QAHE, very scarce works have shown the QAHE in TI/magnetic material heterostructures. Watanabe et al. [111] reported the QAHE in Zn_1−x_Cr_x_Te/(Bi_y_Sb_1−y_)_2_Te_3_/Zn_1−x_Cr_x_Te but only at 100 mK, which is much smaller than the T_c_ of the adjacent Zn_1−x_Cr_x_Te at 60 K. The major challenge of high-temperature QAHE in TI/magnetic material heterostructures is the band-bending effect caused by the interfacial charges [112], as illustrated in Figure 6d. The band gap of commonly used magnetic insulators is larger than the bulk band gap of TIs, which leads to band bending on the TI surface. This ensures that the tunnelling of surface state electrons to the bulk valence band of TI is possible at elevated temperatures.

## 6. Conclusions and Outlooks

MTI is a fertile playground to investigate various topological phases. In summary, we review the main MTI materials and possible topological phases that can exist in MTIs, as well as the scaling behavior during the topological phase transitions in mesoscopic MTIs. To switch between different topological phases, pressure modulation, electrical field tuning, and proximity effect-induced interfacial engineering are discussed as effective tools to navigate through material parameter spaces. Down the road, we anticipate a few research focal frontiers in the field of MTI materials which need to be addressed as priorities to further explore the interactions between magnetism and topology and bring MTI into practical use.

Improving QAH temperatures. The known MTIs suffer from material quality such as inhomogeneous doping and surface quality. Current state-of-the-art MTIs can only achieve the QAH effect at around 2 K, which is substantially lower than the calculated topological gap and experimentally observed Curie temperature. Tensile strain seems to be a promising route to improve the QAH temperature [77]. Advances remain to be explored by either optimizing material preparation or tuning the material properties with external conditions such as strain, electric field, and other parameter spaces.

Material prediction. With the recent rapid development of machine learning-powered high-throughput material search [113,114,115,116], new types of MTIs can be expected with more desirable properties, accessible temperatures, and more exotic physics. In addition, theoretical predictions on proximitizing topological insulators with layered 2D magnets are also expected to be an active field in the theoretical frontier.

Device applications. With the merit of dissipation-less transport and field-free operations, the QAH effect has been widely proposed for application in various devices, for example, spintronic devices [117,118,119,120], neuromorphic devices for high-performance in-memory computing [121], and cryogenic devices for quantum applications [122,123]. It should be acknowledged that the current state-of-the-art QAH materials can only be operated in the sub-Kelvin temperature range as limited by the material quality; therefore, the utilities are restricted to specific application domains such as quantum computing and cryogenic electronics. With the continuous improvement of material quality, we expect that more general and versatile applications of MTIs will be achieved in the future.

## Figures and Tables

**Figure 1 nanomaterials-13-02655-f001:**
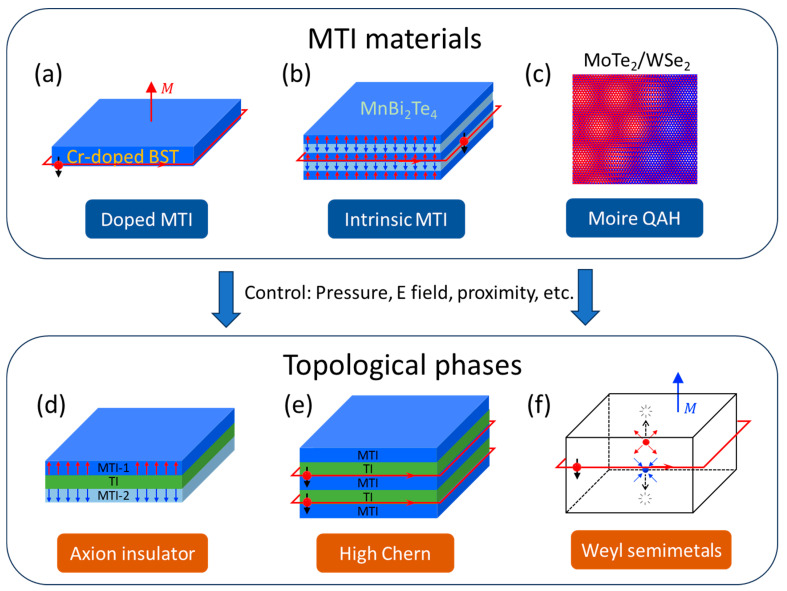
**Overview of magnetic topological insulators (MTIs) and accessible topological phases.** (**a**) Cr-doped (Bi,Sb)_2_Te_3_ is the most well-developed MTI material that hosts quantum anomalous Hall effect with topologically protected chiral edge states. (**b**) An example of intrinsic MTIs, MnBi_2_Te_4_, which features a layered antiferromagnetic structure. (**c**) Orbital magnetism makes twisted Moiré superlattice (e.g., MoTe_2_/WSe_2_) a new type of MTI. (**d**) Axion insulator states with opposite magnetic moments (blue and red arrows) of top and bottom surfaces. (**e**) MTI/TI superlattice with parallel chiral edge states (with unidirectional current as indicated by red arrows) and high-Chern-number insulating states. (**f**) Magnetic Weyl semimetals feature a pair of Weyl nodes (with diverging or converging Berry Curvature as indicated by the blue and red arrows) and evolve into QAH states when approaching the 2D limit.

**Figure 2 nanomaterials-13-02655-f002:**
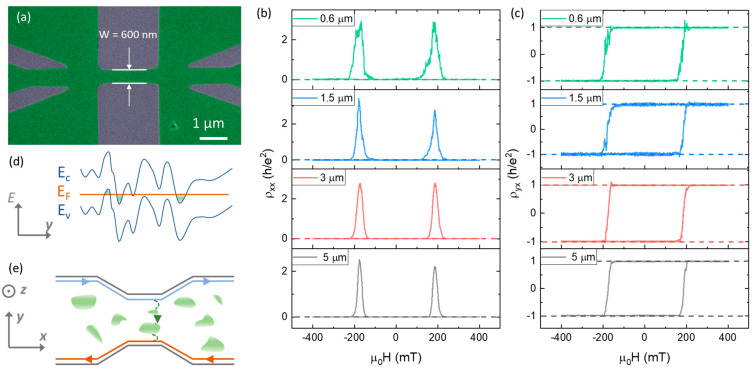
**Mesoscopic transport and QAH effect in sub**–**µm**–**sized MTI samples.** (**a**) An SEM image of a miniaturized MTI Hall bar device with a width of 600 nm. (**b**,**c**) QAH behavior in (**b**) $\rho_{xx}$ and (**c**) $\rho_{xy}$ in various sizes ranging from 5 µm to 600 nm. (**d**) Energy profiles and fluctuations due to dislocations and inhomogeneity in the film. (**e**) Schematics of 2D puddles-assisted back-scattering mechanism. (**a–c**) are adopted from ref [56].

**Figure 3 nanomaterials-13-02655-f003:**
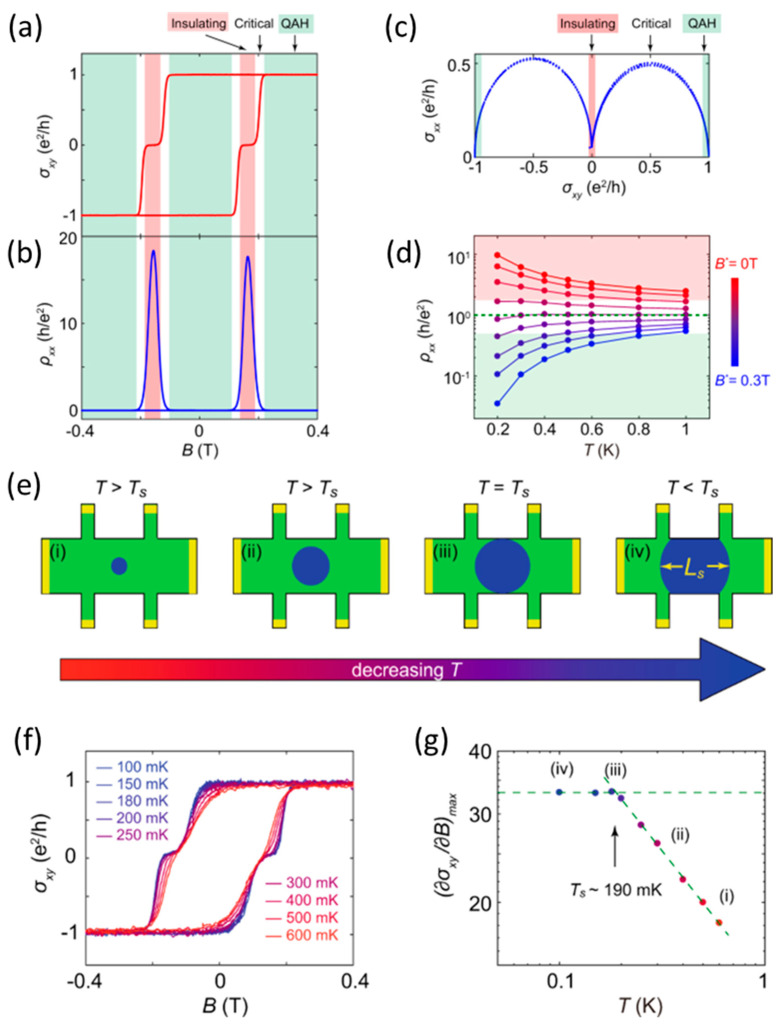
**Scaling behavior during topological transitions in mesoscopic MTI devices.** (**a**,**b**) Magnetic hysteresis loop of (**a**) $\rho_{xy}$ (red) and (**b**) $\rho_{xx}$ (blue) with different topological regimes. (**c**) Semicircle plot of $\rho_{xy}$ and $\rho_{xx}$ marks the topological phase transition. (**d**) Magnetization–driven metal–to–insulator transitions. (**e**) Demonstrations of finite–size effect with increasing inelastic scattering length at lower temperatures. (**f**) Quantized $\rho_{xy}$ at different temperatures in a 5–µm–wide MTI device. (**g**) Scaling behavior of the mesoscopic MTI device with a saturation temperature below which the scaling is dominated by the size limit of the sample. (Two dashed lines are guides to the eyes) (**a**–**f**) are adapted from ref. [62].

**Figure 4 nanomaterials-13-02655-f004:**
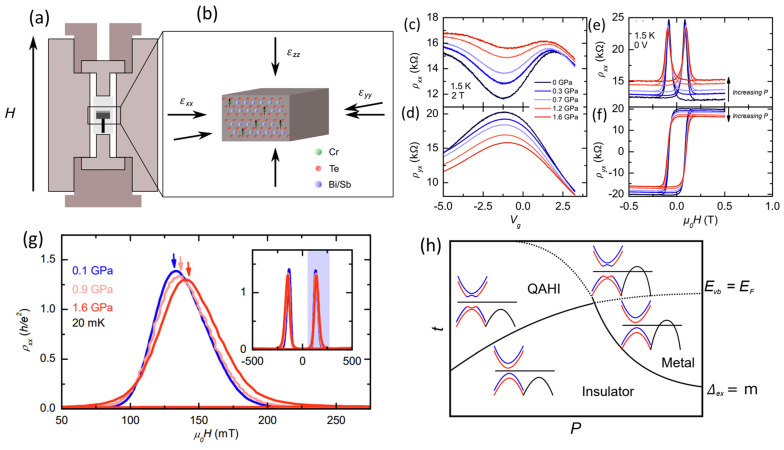
**Pressure tuning of topological states in CBST.** (**a**) Setup of a piston cell that provides hydrostatic pressure for low-temperature magneto-transport measurement. (**b**) Schematics of tri-axial pressure on CBST unit cell. (**c**,**d**) Gate sweep trace of longitudinal resistance (**c**) and transverse resistance (**d**) under different pressures. (**e**,**f**) Magnetic hysteresis loop of longitudinal resistance (**e**) and transverse resistance (**f**) under different pressure. (**g**) Enhanced magnetism with increasing pressure, with arrows marking the coercive fields under different presures. (**h**) Proposed topological phase diagram as a function of film thickness t and pressure P. The phase transition is dictated by two critical conditions: when valence band energy reaches the Fermi level (E_vb_ = E_f_), and when the exchange gap matches the hybridization gap ($\Delta_{ex}=m$). (**a**–**d**) are adapted from ref. [77].

**Figure 6 nanomaterials-13-02655-f006:**
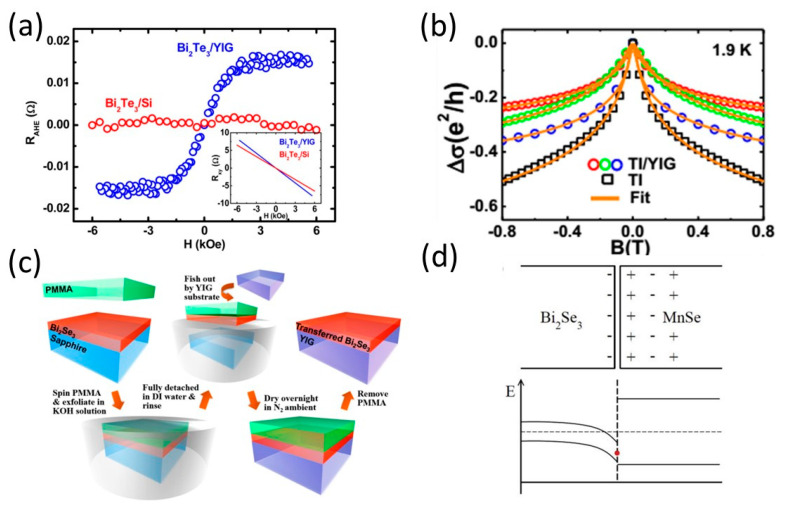
**Inducing magnetic topological properties through magnetic proximity effect.** (**a**) Proximity-induced ferromagnetism in Bi_2_Te_3_/YIG bilayers at 2 K where there is no observed anomalous Hall signal at reference Bi_2_Te_3_ sample. (**b**) Low-field magneto-conductance on Bi_2_Se_3_/YIG and Bi_2_Se_3_ at 1.9 K. Weak anti-localization effect in Bi_2_Se_3_ is suppressed due to the proximity-induced ferromagnetism that breaks the time-reversal symmetry in TI. (**c**) Schematic of the procedures in transferring an MBE-grown Bi_2_Se_3_ film onto a YIG substrate via wet transfer. (**d**) Charge states at the Bi_2_Se_3_/ferromagnetic insulator MnSe interface (top), and band diagram of the junction between the Bi_2_Se_3_ and MnSe, indicating the band bending. The red dot at the interface represents the surface state Dirac point. Figure 6a is adapted from ref. [107], Figure 6b from ref. [106], Figure 6c from ref. [109], and Figure 6d from ref. [112].

**Table 1 nanomaterials-13-02655-t001:** Comparison between the electric field-controlled topological phases and their one-dimensional topological edge states in different quantum materials.

Materials	Topology	∆ (meV)	E_C_ (V/nm)
InAs/GaSb	2D TI	3–9	~0.1 ^†^ [97]
Na_3_Bi	2D TI	~300	1.12 [98]
Graphene/WSe_2_	2D TI	~0.25	0.02 [99]
h-BiSbTeSe_2_	3D TI	~16	~0.25 [100]
(Cr,Bi,Sb)_2_Te_3_	QAHI	~0.04 *	0.9 [40]
tMBG	Orbital CI	~0.4	~0.45 [101]
MoTe_2_/WSe_2_	QAHI	~1.6	0.69 [18]

^†^ The critical electric fields in InAs/GaSb quantum wells are thickness- and composition-dependent. The value here is based on a 50/5/12.5/50 nm AlSb/InAs/GaSb/AlSb double quantum well as reported in ref. [97]. * Gap size could be underestimated due to spatial potential fluctuation in magnetically doped TIs.

## Data Availability

No new data were created or analyzed in this study. Data sharing is not applicable to this article.

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
