# Peer review of "Manipulating Topological Phases in Magnetic Topological Insulators"

_nanomaterials, 2023, doi:10.3390/nano13192655_

Round 1
Reviewer 1 Report
The manuscript is a review paper devoted to tuning topological phases in magnetic topological insulators by applying the finite size effect, hydrostatic pressure, electric field, and the magnetic proximity effect. The paper is interesting and valuable for the further development of these topological materials.
I have only one comment. In section 4.3, the authors introduced a parameter called the critical electric field (Ec), which is required to induce the topological phase transition in various material systems. In Table I, the values of Ec are compared for several materials. However, the Ec is well defined for a particular sample, and not for a material system. For different samples of the same material system (for example InAs/GaSb quantum wells), the values of Ec can be very different. Therefore, the Ec is not a good parameter for comparing different material systems.
I will recommend this manuscript for publication in Nanomaterials when the authors clarify the problem of the Ec.
Author Response
The authors sincerely appreciate the editor’s support and the reviewers’ constructive comments on our manuscript. We have carefully studied the referee’s valuable suggestions and hence revised our manuscript to address these comments accordingly. We believe that these changes have significantly improved the overall quality of the manuscript, and hope that the revised version has met the high standards of Nanomaterials and can be accepted for publication.
Here below are the point-to-point responses to reviewers’ comments.
Reviewer #1 (Remarks to the Author):
The manuscript is a review paper devoted to tuning topological phases in magnetic topological insulators by applying the finite size effect, hydrostatic pressure, electric field, and the magnetic proximity effect. The paper is interesting and valuable for the further development of these topological materials.
I have only one comment. In section 4.3, the authors introduced a parameter called the critical electric field (Ec), which is required to induce the topological phase transition in various material systems. In Table I, the values of Ec are compared for several materials. However, the Ec is well defined for a particular sample, and not for a material system. For different samples of the same material system (for example InAs/GaSb quantum wells), the values of Ec can be very different. Therefore, the Ec is not a good parameter for comparing different material systems.
I will recommend this manuscript for publication in Nanomaterials when the authors clarify the problem of the Ec.
Response: The authors appreciate the reviewer’s insightful suggestion and understand the concern of using critical electric field to benchmark different TI systems. In the literature, dual-gated device structures are typically used, allowing separate control of carrier density and electric field. Therefore, the electric field (rather than gate voltage) can be pinpointed as a direct cause of topological phase transitions. If one considers the fundamental origin of these band evolutions under dual-gate modulation, it is typically attributed to the Stark effect induced band closing and reopening, which is also fundamentally rooted in the electric field modulation.
Alternatively, the top and bottom surface states hybridization is proposed to explain the topological phase transition in InAs/GaSb quantum wells. In this case, we acknowledge that the chemical potential difference across the quantum well rather than the electric field is attributed to the topological phase transition, and indeed, as the reviewer pointed out, the Ec is subject to the width of the quantum well and is device specific. However, given that the window of the allowed width of the quantum well to form 2D TI is limited, the Ec will not differ drastically. Hence we argue that the critical electric field is a parameter, at least within a reasonable range, to benchmark the required condition to achieve topological phase transition.
Reviewer 2 Report
The paper is a short review on the materials combining the complex topology and introduced magnetism. These materials may occur in different topological phases depending on tuning material parameters such as doping profiles, interfacial proximity effect, or external conditions such as pressure and electric field. The main role in the review plays the quantum anomalous Hall (QAH) effect and conditions how it can be achieved. The interplay between topology and magnetism in magnetic topological insulators is discussed in details. The topic of the paper is modern and attracts much attention. The paper is written by the manner to underline the physical nature of discussed phenomena. In the time being QAH can be only achieved at ~2K, which is substantially lower than the calculated topological gap and experimentally observed Curie temperature. The authors links this discrepancy with inhomogeneous magnetic doping and surface imperfections. Ways how to increase the possible QAH temperature is one of the most valuable purposes of the paper. External strain, electric field, and other parameter spaces are regarded as the tuning parameters. The authors predict that new types of magnetic topological insulators can be obtained with more desirable properties, accessible temperatures, and more exotic physics. The paper falls into the scope of the journal. Conclusion of the paper is mostly supported by the text. The list of references is appropriate. The paper can be published after minor corrections.
1. The authors in Conclusion and Outlooks discuss the device applications of materials with QAH effect. Taking into consideration the merit of dissipation-less transport and field-free operations, it is, however, worthy to notice that because of very low temperature for QAH effect realization, applications seems to be evidently limited.
2. Misprint in p.9, line 4 from the top.
3. The capture to Fig.5. Please, point out the ref. from which Fig. 4c and 4d are taken.
4. Please, correct invalid citation 83.
Author Response
Response Letter
The authors sincerely appreciate the editor’s support and the reviewers’ constructive comments on our manuscript. We have carefully studied the referee’s valuable suggestions and hence revised our manuscript to address these comments accordingly. We believe that these changes have significantly improved the overall quality of the manuscript, and hope that the revised version has met the high standards of Nanomaterials and can be accepted for publication.
Here below are the point-to-point responses to reviewers’ comments.
Reviewer #2 (Remarks to the Author):
The paper is a short review on the materials combining the complex topology and introduced magnetism. These materials may occur in different topological phases depending on tuning material parameters such as doping profiles, interfacial proximity effect, or external conditions such as pressure and electric field. The main role in the review plays the quantum anomalous Hall (QAH) effect and conditions how it can be achieved. The interplay between topology and magnetism in magnetic topological insulators is discussed in details. The topic of the paper is modern and attracts much attention. The paper is written by the manner to underline the physical nature of discussed phenomena. In the time being QAH can be only achieved at ~2K, which is substantially lower than the calculated topological gap and experimentally observed Curie temperature. The authors links this discrepancy with inhomogeneous magnetic doping and surface imperfections. Ways how to increase the possible QAH temperature is one of the most valuable purposes of the paper. External strain, electric field, and other parameter spaces are regarded as the tuning parameters. The authors predict that new types of magnetic topological insulators can be obtained with more desirable properties, accessible temperatures, and more exotic physics. The paper falls into the scope of the journal. Conclusion of the paper is mostly supported by the text. The list of references is appropriate. The paper can be published after minor corrections.
- The authors in Conclusion and Outlooks discuss the device applications of materials with QAH effect. Taking into consideration the merit of dissipation-less transport and field-free operations, it is, however, worthy to notice that because of very low temperature for QAH effect realization, applications seems to be evidently limited.
Response: The authors appreciate the reviewer’s positive comment on our manuscript. We agree with the reviewer that the current state-of-the-art QAH materials can only be operated in the sub-Kelvin temperature range as limited by the material quality, therefore the utilities are restricted to specific application domains such as quantum computing and cryogenic electronics. While it is challenging for universal application scenarios at this stage, proof-of-concept device has been demonstrated, such as cryogenic in-memory computing devices (ref 121), miniaturized microwave circulators for quantum computing (ref 122, 123), and spin-orbit torque switching devices. With the continuous improvement of material quality, we expect more general and versatile applications will be achieved in the future.
In the revised manuscript, in the Outlooks section we addressed the current challenges with low operating temperature:
It should be acknowledged that the current state-of-the-art QAH materials can only be operated in the sub-Kelvin temperature range as limited by the material quality, therefore the utilities are restricted to specific application domains such as quantum computing and cryogenic electronics. With the continuous improvement of material quality, we expect more general and versatile applications with MTIs will be achieved in the future.
- Misprint in p.9, line 4 from the top.
Response: the sentence has been rephased for clarity:
“However, when the physical size of the device is below certain characteristic length of the electron quantum transport, (i.e. phase coherence length), the electron transport behavior quickly deviates from the classical diffusive regime and should be characterized by coherent mesoscopic transport.”
- The capture to Fig.5. Please, point out the ref. from which Fig. 4c and 4d are taken.
Response: The refeence for Fig. 5c and 5d has been updated (ref. 90) in the revised manuscript.
- Please, correct invalid citation 83.
Response: The reference 83 has been corrected:
83 Li, J. & Wu R., Electrically Tunable Topological Phase Transition in van der Waals Heterostructures. Nano Letters 23, 2173 (2023).
Round 2
Reviewer 1 Report
The response of the authors to my comment in the previous review is not satisfactory. The parameter Ec remains unclear, particularly for the InAs/GaSb quantum well (QW) system. In this QW system, the topological phase transition originates from the fact that the valence band edge of GaSb is higher than the conduction band edge of InAs. Bilayer InAs/GaSb QWs and three-layer InAs/GaSb/InAs QWs can be in the normal insulator phase or in the topological insulator phase or in the topological phase transition (Dirac semimetal or Weyl semimetal), depending on the widths of InAs and GaSb layers (see Liu et al., Phys. Rev. Lett. 100, 236601 (2008), and Krishtopenko and Teppe, Sci. Adv. 4, eaap7529 (2018) ). The electric field is not needed to obtain an InAs/GaSb QW in the Dirac/Weyl semimetal state. A similar situation is for HgCdTe/CdTe QWs.
In section 4.3, the authors claim that the parameter Ec determines the electric field that is required to induce the topological phase transition. They estimated the value of Ec for the InAs/GaSb QW system to be about 0.1 V/nm. This result is probably correct for the sample studied in Ref. 97, but not for the entire InAs/GaSb QW system. For HgCdTe/CdTe QWs, the critical gate voltage (equivalent to the Ec) was studied theoretically by Li and Chang, Appl. Phys. Lett. 95, 222110 (2009). It is clear from this work that the critical gate voltage for HgCdTe/CdTe QW system is not a single number but a function of the QW thickness and QW content. This function (see Fig 3. in the paper of Li and Chang) does not saturate and cannot be represented by any single number. Therefore, I am not convinced by the response of the authors that a single value of the Ec= ~0.1 V/nm can reasonably represent the InAs/GaSb QW system.
I do not recommend this manuscript for publication in its present form since the parameter Ec is unclear and may be misleading.
Author Response
Response: The authors thank the reviewer for further clarification on the InAs/GaSb quantum well’s (QW) case. After thoroughly studying the literature the reviewer mentioned, we agree with the reviewer that Ec is not a universal parameter to evaluate the critical behavior of topological phase transition in InAs/GaSb QWs. Fundamentally, the formation of 2D TI inverted bands in InAs/GaSb QWs is different from other intrinsic TIs listed in Table I. In intrinsic TIs, the topological gap is created by the hybridization of bands from different orbitals within a single material platform. In these cases, the electric field directly alters the band structure in certain ways, for example, through Stark effect, or enhanced spin-orbit coupling. Hence for the other intrinsic TIs, it is appropriate to compare the critical field as a parameter for topological phase transition. However, in InAs/GaSb and HgTe/CdTe class QW based class of TIs, the band inversion is created by the band misalignment in the heterostructures. The electric field drives the system into the topological phase transition by inducing chemical potential difference across the QW and changes relative position of E1 and H1 bands. In this case the voltage change may be more suitable for capturing the band misalignment, and indeed, the critical field should be thickness dependent, as the reviewer suggested. To avoid any misinterpretation, in the revised manuscript, we have included an annotation to inform the readers that the critical field in the first line of the table is device-specific:
† The critical electric fields in InAs/GaSb quantum wells are thickness and composition dependent. The value here is based on a 50/5/12.5/50 nm AlSb/InAs/GaSb/AlSb double quantum well as reported in ref. 97.
† The critical electric fields in InAs/GaSb quantum wells are thickness and composition dependent. The value here is based on a 50/5/12.5/50 nm AlSb/InAs/GaSb/AlSb double quantum well as reported in ref. 97.